# The Effects of Insulin-Like Growth Factor I and BTP-2 on Acute Lung Injury

**DOI:** 10.3390/ijms22105244

**Published:** 2021-05-15

**Authors:** Kevin Munoz, Samiksha Wasnik, Amir Abdipour, Hongzheng Bi, Sean M. Wilson, Xiaolei Tang, Mahdis Ghahramanpouri, David J. Baylink

**Affiliations:** 1Department of Medicine, Division of Regenerative Medicine, Loma Linda University, Loma Linda, CA 92354, USA; kmunoz@students.llu.edu (K.M.); swasnik@llu.edu (S.W.); AAbdipou@llu.edu (A.A.); Xiaolei.tang@liu.edu (X.T.); mghahramanpouri@llu.edu (M.G.); 2Division of Nephrology, Loma Linda University Medical Center, Loma Linda, CA 92354, USA; 3Henan Institute of Medical and Pharmaceutical Sciences, Zhengzhou University, Zhengzhou 450052, China; hongzhengbi@zzu.cn; 4The Lawrence D. Longo, MD Center for Perinatal Biology, Department of Basic Sciences, Loma Linda University School of Medicine, Loma Linda, CA 92354, USA; seanwilson@llu.edu; 5Department of Veterinary Biomedical Sciences, College of Veterinary Medicine, Long Island University, Brookville, NY 11548, USA

**Keywords:** acute lung injury, inflammation, calcium channels, cytokines, LPS, vascular integrity

## Abstract

Acute lung injury (ALI) afflicts approximately 200,000 patients annually and has a 40% mortality rate. The COVID-19 pandemic has massively increased the rate of ALI incidence. The pathogenesis of ALI involves tissue damage from invading microbes and, in severe cases, the overexpression of inflammatory cytokines such as tumor necrosis factor-α (TNF-α) and interleukin-1β (IL-1β). This study aimed to develop a therapy to normalize the excess production of inflammatory cytokines and promote tissue repair in the lipopolysaccharide (LPS)-induced ALI. Based on our previous studies, we tested the insulin-like growth factor I (IGF-I) and BTP-2 therapies. IGF-I was selected, because we and others have shown that elevated inflammatory cytokines suppress the expression of growth hormone receptors in the liver, leading to a decrease in the circulating IGF-I. IGF-I is a growth factor that increases vascular protection, enhances tissue repair, and decreases pro-inflammatory cytokines. It is also required to produce anti-inflammatory 1,25-dihydroxyvitamin D. BTP-2, an inhibitor of cytosolic calcium, was used to suppress the LPS-induced increase in cytosolic calcium, which otherwise leads to an increase in proinflammatory cytokines. We showed that LPS increased the expression of the primary inflammatory mediators such as toll like receptor-4 (TLR-4), IL-1β, interleukin-17 (IL-17), TNF-α, and interferon-γ (IFN-γ), which were normalized by the IGF-I + BTP-2 dual therapy in the lungs, along with improved vascular gene expression markers. The histologic lung injury score was markedly elevated by LPS and reduced to normal by the combination therapy. In conclusion, the LPS-induced increases in inflammatory cytokines, vascular injuries, and lung injuries were all improved by IGF-I + BTP-2 combination therapy.

## 1. Introduction

Acute lung injury (ALI) often manifests as acute respiratory failure. ALI can occur from local or systemic causes and can result from infectious or noninfectious causes [1]. Acute lung injury and acute respiratory distress syndrome (ARDS) are considered a continuum [1]. It has been estimated that ~200,000 patients are afflicted annually, with a 40% mortality rate [2]. However, with the pandemic of COVID-19, ALI is massively increased. ALI therapy is currently mainly supportive, although new therapies are rapidly emerging [3]. Pathologically, both the alveolar epithelium and the vascular endothelium are injured in ALI [4]. Additionally, there is a large increase in the serum and local expression of proinflammatory cytokines, particularly tumor necrosis factor-α (TNF-α), interleukin-1β (IL-1β), interleukin-6 (IL-6), interleukin-8 (IL-8), and interleukin-18 (IL-18) [5]. It has been suggested that the severity of ALI is influenced by neutrophil migration into the lungs in response to activated alveolar macrophages [6].

Our goal was to develop an ALI therapy in a mouse model induced by lipopolysaccharide (LPS). LPS is one of the several approaches to create experimental ALI [7]. Often, mouse results do not translate to humans. However, in support of the use of LPS for human ALI, it has been shown that the small primate, marmosets, develop ALI in response to LPS [8]. 

To treat LPS-induced ALI, we elected to utilize the following two therapeutic agents: insulin-like growth factor-I (IGF-I) and the store-operated calcium channel inhibitor (BTP-2). IGF-I therapy was selected, because we observed a reduction in the serum IGF-I in our past sepsis studies [9]. Previously, we found that the decrease in serum IGF-I during sepsis was associated with the depression of growth hormone receptor (GHR) expression in the liver and increased circulating inflammatory cytokines [9]. Since inflammatory cytokines were known to suppress GHR expression in the liver, which was the primary source of circulating IGF-I, we reasoned that the suppressed liver GHR expression accounted for the low-serum IGF-I under systemic inflammatory conditions. 

Several IGF-I functions described from previous studies suggest that it is critical to normalize IGF-I during ALI. Firstly, IGF-I has been shown to enhance the proliferation and differentiation of lung epithelial cells. Secondly, IGF-I can modulate immune cells to reduce proinflammatory cytokine production [10]. Thirdly, with systemic LPS administration, there is usually a vascular injury component [11]. In this regard, IGF-I has several functions that could counteract the harmful effects of LPS on the vascular function [12,13]. IGF-I increases the expression of endothelial adhesion molecules, the proliferation of the resident endothelial progenitor cells, and the 25-hydroxyvitamin D, 1α hydroxylase activity that produces the active metabolite of vascular-protective vitamin D [14]. Fourthly, IGF-I also activates the endoplasmic reticulum (ER) calcium ATPase (SERCA), which is responsible for the transfer of calcium from the cytosol to the ER [15] and, thus, lowers the cytosolic calcium level. The above-mentioned IGF-I function is relevant to our LPS model, which exhibits an increase in cytosolic calcium [16]. Finally, IGF-I is a pleiotropic growth factor and has been shown to improve lung tissue repair [17].

BTP-2 was selected for treating the LPS-induced inflammation, because it is a specific inhibitor of Orai1 and 2. In this regard, Orai1 and 2 form the core of the calcium release-activated channel (CRAC), promoting a calcium influx across the plasma membrane [18,19]. Thus, the BTP-2 treatment will decrease the cytosolic calcium. The CRAC channel is part of the store-operated calcium entry complex (SOCE), whereby changes in the ER calcium regulate calcium entry through the plasma membrane from the interstitial fluid [20]. IGF-I and BTP-2 both influence the cytosolic calcium levels but by different mechanisms. The increase in cytosolic calcium by LPS is a significant event in terms of creating tissue damage. One of the critical consequences of an increase in cytosolic calcium is that it substantially contributes to the enhanced inflammatory cytokine production [21]. 

LPS initiates its cellular toxicity by binding to and activating the TLR-4 complex, which leads to an increase in cytosolic calcium. The increase in cytosolic calcium mediates many of the adverse effects of LPS on the cellular functions [22]. There are several potential inhibitors of SOCE [23]. However, we chose BTP-2, because it has been used successfully in experimental animals to suppress the immune system in severe inflammatory conditions, including ALI [24]. The immunosuppressive activity of BTP-2 is similar to cyclosporine, but cyclosporine is reportedly more toxic. Thus, BTP-2 can block the calcium-dependent major LPS-induced detrimental pathways. 

In the current study, we examined the effects of the IGF-I and BTP-2 therapies, individually and in combination, in an LPS mouse model of ALI (Figure 1). The present ALI study builds on our previous work, in which we examined a mouse model of kidney pathogenesis by systemic LPS administration [13]. We now examined the lung pathology and treatment from the same animals, comparing our findings in the kidneys. This study approach is advantageous, because it allowed us to compare, in the same animals, the effects of the IGF-I and BTP-2 therapies on two significant organs that sustained LPS-induced inflammatory injury. 

## 2. Results

### 2.1. Past Work on Serum IGF-I Directly Applicable to the Current Study

Our past work on the animals used in this current study showed that the serum IGF-I level was significantly reduced in the animals injected with LPS [13]. Therefore, a lentiviral vector engineered to overexpress IGF-I was injected into the skeletal muscle to produce a stable average level of serum IGF-I and rescue the IGF-I deficiency. We reasoned that the constant IGF-I production would be required to maintain a normal serum IGF-I, because, being a small protein, IGF-I would be expected to have a shorter half-life [25,26]. Under physiological conditions, growth hormone acts on the GHR in the liver to stimulate IGF-I production and large IGF-binding proteins (IGFBPs). The binding of IGFBPs to IGF-I protects IGF-I from rapid excretion. However, in the LPS-treated animals, elevated proinflammatory cytokines negatively regulate the liver’s GHR gene expression, which would decrease both IGF-I and the IGFBP. Our therapy is only IGF-I without IGFBPs. Therefore, we would expect a short half-life of the muscle-derived IGF-I [27]. 

Clinically, IGF-I is frequently administered subcutaneously; however, this administration route does not result in stable serum IGF-I levels [28]. Serum-free IGF-I is readily available to injured organs, such as the lungs. In this regard, our previous results showed that the in vivo gene therapy rescued the IGF-I deficiency by increasing the serum IGF-I levels to 400 ng/mL as compared to 80 ng/mL in the nontreated LPS-injected mice [13].

### 2.2. Current Studies on the Lung

#### 2.2.1. Intracellular Signaling Pathways

LPS initiates its action by activating the toll like receptor-4(TLR-4)-mediated intracellular signaling pathway [22,29]. In addition, it also increases the expression of the TLR-4 protein [29]. We found a marked increase in the TLR-4 gene expression in the lungs seven days after LPS administration (Figure 2). IGF-I, BTP-2, or the combination therapy decreased the TLR-4 gene expression to the normal level. Based on a previous work [13], TLR-4 signaling reduces the calcium levels in the ER stores via the activation of inositol triphosphate (IP3) signaling [22], which, in turn, increases cytosolic calcium through the release of ER stores and the coordinated activation of the SOCE/CRAC channels [30]. As expected, there was a significant decrease in Orai1 in the BTP-2 and combination therapy groups (Figure 2). The BTP-2 treatment also decreased the gene expression of calcineurin and nuclear factor of activated T cells (Nfat), reflecting a decrease in the cytosolic calcium (Figure 2). In this regard, the BTP-2 treatment also decreased transient receptor potential channel 3 (TRPC3) and TRPC6, known to increase the plasma membrane calcium influx [31,32,33]. These findings suggest that BTP-2 inhibited the expression of Orai1, TRPC3, and TRPC6 and led to a downstream decrease in calcineurin and Nfat (Figure 2). Similar to the BTP-2 treatment, IGF-I also decreased Orai1 and TRPC3 and TRPC6 (Figure 2). The activation of TLR-4 further induces the activation of the downstream signaling pathways, such as mitogen-activated protein kinase (MAPK) and nuclear factor- kB (NF-kB) [34]. In this regard, we found a significant decrease in the gene expression of MAPK and NF-kB in the BTP-2 and IGF-I combination treatment group.

A previous work showed that IGF-I regulates ER calcium handling [15,35]. However, to our knowledge, this is the first evidence of IGF-I affecting the plasma membrane calcium channels. Despite the finding that BTP-2 and IGF-I both decreased Orai1 and the TRPC in our LPS-induced ALI mouse injury model, we found no evidence of additive or synergistic effects between these treatments. Resolving the mechanism and significance of the actions of IGF-I on calcium signaling will require further studies.

#### 2.2.2. Inflammatory Cytokine Gene Expression in the Lungs

LPS administration caused a highly significant increase in the gene expression of the important proinflammatory cytokines IL-1β, IL-6, interleukin-17 (IL-17), interferon-γ (IFN-γ), and TNF-α after seven days of treatment (Figure 3). In sharp contrast, the treatment with IGF-I or BTP-2 or the combination therapy decreased the gene expression of most of the proinflammatory cytokines to normal (Figure 3).

The above-observed effects of IGF-I or BTP-2, or the combination therapy, have important clinical implications. IL-1β is a proinflammatory cytokine produced by many cell types but, particularly, by activated macrophages [36] and functions in general to promote cell proliferation and apoptosis [37]. 

IL-6, produced by mesenchymal cells and immune cells, is a pleiotropic inflammatory cytokine responsible for acute protein synthesis and neutrophil migration in inflammatory sites. Importantly, IL-6 is also required for combating viral infections. IGF-I’s effect to markedly decrease the IL-6 expression in inflammation has not been previously reported, except in our previous study on acute kidney injury (AKI) [13]. It will be essential to determine if the change in IL-6 is calcium-dependent, given our discovery of the IGF-I effects on the ORAI and TRP channels mentioned above. 

Both IGF-I and BTP-2 markedly decreased the IL-17 expression. IL-17 is produced by CD4 T-helper cells and functions as the immune system’s cytokine sentinel. IL-17 performs surveillance functions, maintenance of the mucosal barrier integrity, and recruitment of myeloid cells, increasing granulocyte colony stimulating factor (G-CSF) [38]. 

Both the IGF-I and BTP-2 treatments caused marked decreases in the expression of lung TNF-α, which is primarily produced by macrophages [39,40]. TNF-α triggers the production of both IL-1β and IL-6. Significantly, TNF-α adversely affects endothelial adhesion molecules, resulting in vascular leakage. Interestingly, it is known that TNF-α downregulates the hepatic GHR expression, leading to a depression of circulating IGF-I during the inflammatory states [41]. Although our model is one of sterile inflammation, in studies of infectious inflammation, TNF-α and IFN-γ are important proinflammatory cytokines known to create tissue damage. Interestingly, both cytokines’ gene expressions were decreased by the treatment with IGF-I or BTP-2 or the combination therapy. With respect to the expression of IFN-γ, which the IGF-I and BTP-2 therapies corrected, it has been shown that IFN-γ-deficient mice are resistant to septic shock [42]. 

#### 2.2.3. Vascular Integrity

A quantitative gene expression analysis was done to measure the CD31 (platelet endothelial cell adhesion molecule-1 or PECAM-1), α-smooth muscle actin (α-SMA), and vascular endothelial growth factor (VEGF) expressions relevant to endothelial cells. LPS caused a significant decrease in the VEGF gene expression in the lungs (Figure 4). BTP-2 therapy markedly increased the VEGF expression, whereas IGF-I did not affect the VEGF expression. The physiological significance of the VEGF expression changes is unclear, as VEGF is also associated with vascular leakage. Accordingly, VEGF expression changes could represent a repair process or a potential mechanism for the observed leakage [43,44].

The IGF-I and BTP-2 combination therapy did not increase the VEGF expression, raising the possibility that IGF-I inhibited the effect of BTP-2 on the VEGF expression. CD31, another gene related to vascular integrity, was markedly decreased by LPS but corrected by BTP-2 and IGF-I therapy and the combination therapy (Figure 4). The immunocytochemical evaluation of CD31 on the lung vasculature revealed that the amount of space occupied by endothelial cells was markedly reduced by LPS administration and significantly improved by IGF-I, BTP-2, or the combination therapy (Figure 4). Thus, the gene expression of CD31 and the immunocytochemical evaluation of CD31-positive cells showed an improvement with all three therapeutic groups.

Specific to the vasculature is α-SMA, a marker for smooth muscle actin, but it can be an index of pericyte activity [45,46]. Pericytes are located at endothelial cell junctions and function to prevent vascular leakages [47]. LPS caused a significant decrease in the α-SMA gene expression and α-SMA^+^ cells (Figure 4). IGF-I therapy completely corrected the LPS-induced reduction in α-SMA gene expression, whereas BTP-2 had no beneficial effects. The combination therapy reflected the impact of BTP-2 rather than IGF-I. However, the immunohistochemistry analyses showed that all three therapeutic modalities ameliorated the LPS-induced decrease in the α-SMA^+^ cells (Figure 4B).

### 2.3. Lung Tissue Damage and Repair

The synthesis of antimicrobial protein neutrophil gelatinase-associated lipocalin-2 (LCN2/NGAL) increases in bronchial epithelial cells in response to inflammation [48]. The NGAL expression was significantly increased in the lungs in response to LPS (Figure 5). Both the IGF-I and BTP-2 therapies significantly reduced the elevated NGAL expression almost to normal, as did the combination therapy. Surfactant protein D (SP-D) is a marker of alveolar epithelial cell proliferation during acute lung injury [49].

We found a marked decrease in SP-D in response to LPS. Both IGF-I and BTP-2 normalized the marked depression of SP-D in the lungs, as did the combination therapy. The LPS treatment markedly increased the expression of caspase 3. We did not measure the caspase 3 catalytic activity or apoptosis; however, LPS is known to cause endothelial cell apoptosis [50,51,52]. With respect to the treatments with either IGF-I or BTP-2, both types of therapies corrected the elevated caspase 3 expression produced by LPS, as did the combination therapy (Figure 5). 

#### Quantitative Lung Histology

Next, we supplemented the epithelial injury gene expression studies with four quantitative histological analysis parameters of the lung injury. LPS, at seven days, showed a marked disruption of the lung epithelium, as is evident from the quantitative data below and a massive invasion of inflammatory immune cells (Figure 6 and Figure 7).

Mean linear intercept: An increase in the mean linear intercept indicates an abnormal increase in the airspace. The mean linear intercept is the free distance between the gas exchange surfaces in the acinar airway complex. The mean linear intercept was increased in the LPS group, indicating an abnormal increase in the airspace, leading to a reduced gas exchange. All three treatment modalities improved the mean linear intercept score (Figure 7A).

Destructive index: The destructive index is a measure of the alveolar septal damage. The criteria are alveolar septal breaks or collapses and airspace enlargement. In the LPS group, there was septal thickening and leukocyte infiltration of the alveoli. Accordingly, the damage was markedly increased in the LPS group and improved in all three therapeutic modalities (Figure 7B). LPS markedly increased the destructive index, whereas the other three therapies normalized this index. 

Area disrupted: The area disrupted parameter is a measurement of the percentage of visible area damage. The damage was indicated by septal thickening and leukocyte infiltration in the alveoli. LPS-induced inflammation caused neutrophil-dependent emphysematous changes in the lung architecture and apoptosis. All three therapeutic modalities improved these changes (Figure 7C). 

Mean septal thickness: The septal thickness represents the blood–air barrier thickness, which increases with an injury. The treatment with LPS caused a significant increase in the mean septal thickness, whereas all three therapeutic modalities normalized this parameter (Figure 7D). 

We integrated the four parameters into a total lung injury score, which showed that all the therapies, including the combination IGF-I + BTP-2 therapy, normalized the lung injury score (Figure 7E).

## 3. Discussion

The current study was undertaken to determine if IGF-I, BTP-2, or IGF-I + BTP-2 therapy would ameliorate the lung damage in a mouse model of LPS-induced ALI. Additionally, the study also evaluated the mechanisms responsible for the favorable therapeutic responses, particularly the role of IGF-I and BTP-2 therapies in cellular calcium regulation. Since the mice utilized in this lung study were also previously evaluated for acute kidney injury [13], we provided a comparison of the therapeutic and mechanistic features of these two major organs.

### 3.1. Assessment of Therapy

The overall approach for the therapeutic analysis of the IGF-I and BTP-2 therapies for LPS-induced ALI was to focus on three essential biologic processes of inflammation and repair—namely, (1) increased production of the proinflammatory cytokines, (2) impaired vascular function, and (3) improved repair (or less injury). The relevant gene expressions and histopathology evaluations determined the IGF-I and BTP-2 beneficial therapeutic effects. IGF-I, BTP-2, and the combination therapy had positive therapeutic actions. We found no instance of synergism between IGF-I and BTP-2. These positive therapeutic findings are consistent with our earlier findings that our therapies were effective in our AKI mouse investigations [13].

### 3.2. Inflammatory Cytokines Gene Expression

In the present study, all three IGF-I, BTP-2, and the combination therapies were remarkably suppressive of the expression of IL-1β, IL-6, IL 17, TNF-α, and IFN-γ adverse inflammatory cytokines as compared to the LPS group. It is important to note that IGF-I was as effective as BTP-2 in suppressing cytokine expression in the lungs. We expected the effectiveness of BTP-2 because of its potent inhibition of Orai1. However, IGF-I is not known to be so remarkably immunosuppressive. These therapeutic changes are similar to those found in the kidney study [13]. However, the decrease in TNF-α gene expression was greater in the lungs (Figure 3) than in the kidneys [13]. TNF-α is known to have direct adverse effects on vascular pericytes, cells that promote vascular integrity [47]. TNF-α and IFN-γ were normalized in the three therapeutic groups. The above finding is intriguing, because among the proinflammatory cytokines, only TNF-α and IFN-γ synergistically promote cell death by pancytosis [42]. This action on cell death would be expected to release damage-associated molecular patterns (DAMPs), which would promote a sterile inflammatory response, and the infectious inflammatory response of the microbes. Significant sources of TNF-α and IFN-γ are neutrophils and macrophages. In this regard, we found a marked invasion of immune cells characterized by a darkly stained nucleus and an eosinophilic cytoplasm, features typical of neutrophils and macrophages, in the animals receiving LPS without therapy.

### 3.3. Vascular Integrity

LPS adversely affected the gene expression of CD31, VEGF, and caspase 3 in the lung tissue and was markedly improved by the three therapeutic groups. The quantitative immunocytochemistry of CD31 and α-SMA showed suppression with LPS and a significant increase in all the therapeutic groups. Thus, the therapies had a favorable effect on the parameters of the vascular integrity in the lungs. The extent of these therapeutic gene expression changes was not as great in the lungs compared to what was observed in the kidneys in our earlier study, where values above the normal were found in some parameters [13]. This differences between the lungs and kidneys could indicate that the lung injury was repaired more rapidly than that of the kidneys. Further studies will be required to determine the pathologic/repair significance of the lung and kidney differences.

### 3.4. Lung Injury/Tissue Repair

In our ALI study, the synthesis of antimicrobial protein neutrophil gelatinase-associated lipocalin-2 (LCN2 or NGAL) was significantly increased in the lungs in response to LPS. In all three therapeutic groups, there was a significant reduction of the elevated NGAL gene expression. SP-D is a marker of alveolar epithelial cell proliferation during acute lung injury. The SP-D gene expression was markedly decreased by LPS and completely corrected by all three therapies. In our detailed quantitative histological study of the lungs, there was a remarkable improvement in all three therapies. These parameters were incorporated into an injury score, which was almost normalized in all three therapeutic groups, particularly the combination therapy. 

### 3.5. Potential Mechanisms for the Favorable Therapeutic Responses

Our study was designed to focus on cytosolic calcium regulation as a significant factor in the damage caused by LPS and the improvement elicited by IGF-I and BTP-2 and the combination therapy. In this regard, the application of BTP-2 was a mechanical advantage, because it inhibits Orai1, which is a major regulator of cytosolic calcium [30,53,54]. LPS causes increased cytosolic calcium in several different cell types, including immune cells, endothelial cells, and epithelial cells [29,55]. Importantly, BTP-2 has been shown to decrease the release of proinflammatory cytokines [18,24,56], indicating that the observed consequences to the BTP-2 treatment are calcium-dependent. However, improvements in tissue damage are not limited to the local actions of BTP-2 on a given tissue. Improvements in tissue damage by BTP-2 therapy could be due to a combination of the reduced proinflammatory cytokine, improved vascular function, or direct actions of BTP-2 on perturbed SOCE in various cells within the organ, including alveolar epithelial cells in the lungs. 

The BTP-2 treatment reduced LPS-induced ALI injury by influencing the cytosolic calcium signaling and regulatory processes. In particular, the BTP-2 treatment normalized LPS increases in Orai1, TRPC3, and TRPC6, an effect that is predicted to stabilize the cytosolic calcium and downstream transcriptional pathways [30,54]. Accordingly, BTP-2 normalized LPS-mediated increases in calcineurin and Nfat1, which are activated downstream from the increases in the cytosolic calcium through ORAI, the TRPC channels, and the associated SOCE pathways [57,58]. Critically, Nfat1 activation is coupled to proinflammatory cytokine production [57,58]. All of the foregoing changes in response to BTP-2 were seen in our AKI study and others’ works [13,18].

IGF-1 is well-regarded for its pleiotropic effects on the cell functions; however, IGF-I’s influence on cellular calcium is less well-known. Our studies found it striking that IGF-1 had many parallel effects on BTP-2, where IGF-I caused a significant decrease in Orai1, TRPC3, and TRPC6, albeit these effects were more modest than those of BTP-2. Such data provided indirect evidence that IGF-I regulates SOCE. The IGF-I treatment also caused downstream changes in calcineurin and Nfat1 similar in magnitude to those seen in BTP-2.

Therefore, it is reasoned that the IGF-I treatment is mediated, at least in part, through regulatory effects on the cytosolic calcium. IGF-I can induce IP3 in skeletal muscle and, therefore, could release ER calcium into the cytosol, an action that could increase cytosolic calcium [59]. Therefore, further studies are required to understand the complex actions of IGF-I in the context of our LPS model.

One exciting mechanism with respect to LPS-induced inflammation is the increase in the TLR-4 complex, which is an initial effect of LPS-mediated toxicity, is decreased to normal after seven days following the BTP-2 therapy, suggesting that the effect of this SOCE inhibitor somehow feeds back into the immune cells to inhibit the stimulation of the inflammatory response once the cytosolic calcium is normal. Consistent with this concept, TLR-4 expression is regulated by calcium-dependent processes involving calmodulin [60].

Our examinations of LPS-mediated ALI and AKI injury were reminiscent of the multiorgan failure issues observed in COVID-19 patients. Recent evidence illustrates that COVID-19-associated multiorgan failure is linked to impaired mitochondrial function and decreases in the tricarboxylic acid (TCA) cycle function [61]. Often, such depression in mitochondrial energy production is coupled to dysregulation of the calcium overload. It will be engaging in the future to determine if our observations and treatment regimens are related to the mitochondrial calcium overload, which is a cause of cell death. 

Although there were similarities between BTP-2 and IGF-1 on the calcium regulatory pathways examined, the pleiotropic nature of IGF-1 is less causal than BTP-2. The results led to the presumption that BTP-2’s actions are mediated through its influence on cytosolic calcium signaling involving the Orai1- and TRPC-dependent pathways. IGF-1, in comparison, in addition to the influence on the calcium-dependent pathways, is known for its protective effects on the tissue in response to injury and its ability to promote damaged tissue repair through a myriad of mechanisms [13,62]. For example, the reparative functions are not necessarily directly related to cytosolic calcium, including protecting endothelial cell repair [63]. Importantly, concerning future inquiries into the mechanistic relevance of IGF-I is the finding that patients with low-serum IGF-I are more at risk for COVID-19 than those with normal serum IGF-I [64]. That IGF-I therapy improves multiple toxic effects of LPS emphasizes the importance and relevance of IGF-I therapy. 

In multiple cell types, including those in the lungs, increased cytosolic calcium increases Nfat, which acts on numerous pathways downstream, in addition to the promotion of proinflammatory cytokines [65]. Accordingly, our focus on cellular calcium is not intended to diminish the potential importance of other LPS-induced mechanisms of tissue damage.

### 3.6. Potential Clinical Relevance (Lung versus Kidney Responses)

We propose two possible clinical conclusions based on our mouse data examining ALI and AKI due to LPS. Firstly, we propose that IGF-I and BTP-2 mono- or combination therapies would be favorable treatments for systemic inflammatory diseases such as sepsis. Secondly, IGF-I is supported by the FDA for diseases like IGF-I responsive dwarfism [66]. Therefore, IGF-I should be considered in the future as a potential therapy for systemic inflammation under the conditions in which serum IGF-I is decreased. Lastly, BTP-2 is presently a well-described pharmacological research tool but not an FDA-approved drug. Thus, the road for the clinical use of this calcium channel inhibitor will require further preclinical and clinical studies before its use for treatment can be realized. 

## 4. Materials and Methods

### 4.1. Experimental Design

The experimental design is shown in Figure 1.

### 4.2. Animals

Female C57/BL6 mice were purchased from The Jackson Laboratory (Bar Harbor, ME, USA). All mice were used at ages 5–8 weeks. The investigators adhered to the Animal Welfare Act Regulations and other Federal statutes relating to animals and experiments involving animals and the principles outlined in the current version of the Guide for Care and Use of Laboratory Animals, National Research Council. All experiments were performed following the protocols approved by the Institutional Animal Care and Use Committee at Loma Linda University. 

### 4.3. Acute Lung Injury Induced by LPS

Female C57BL/6 mice received a sublethal or lethal dose of LPS from *Escherichia coli* (0127: B8 strain, Sigma Aldrich, St. Louis, MO, USA). LPS in sterile PBS was administered intraperitoneally at 20- or 25-mg/kg body weight.

### 4.4. Treatment

Lenti-IGF-I was administered intramuscularly 24 h before LPS injection. Whereas the TRP and ORAI calcium channel blocker BTP-2 (YM-58483 16 mg/kg, Cayman Chemical, Ann Arbor, MI, USA) was given IP 1 h [31,32] before LPS injection. Some of the animals received a combination of lenti-IGF-I and BTP-2. Mice in control groups were injected with an equal volume of PBS.  

### 4.5. Measurements of mRNA Expression in the Lungs

For the measurement of the mRNA expressions of the whole lung tissues, we performed RT-qPCR. Lung tissues were snap-frozen in liquid nitrogen. According to the manufacturer’s instructions, total RNA was isolated using the RNeasy Micro Kit^®^ (Qiagen, Valencia, CA, USA). First-strand cDNA was synthesized using the SuperScript^®^ III Reverse Transcriptase (Life Technologies, Grand Island, NY, USA). Quantitative RT-qPCR was performed and analyzed in an Applied Biosystems 7900HT Real-Time PCR machine (Applied Biosystems, Foster City, CA). The PCR condition was 10 min at 95 °C followed by 40 cycles of 10 s at 95 °C and 15 s at 60 °C. The relative amount of mRNA was calculated using the comparative *C*_t_ (ΔΔ*C*_t_) method. All specific amplicons were normalized against GAPDH. See Appendix A for a list of the primers.

### 4.6. Lung Immunocytochemistry

After the mice were sacrificed, one part of the lung was immediately cut, fixed in a 10% neutral-buffered formalin solution, embedded in paraffin, and used for histopathological examination. Ten-micrometer-thick sections were cut, deparaffinized, and hydrated. All lung serial sections were incubated at 4 °C overnight with one of two antibodies: rabbit polyclonal antibody against PECAM (CD31) (Santa Cruz Biotechnology, Dallas, TX, USA) or rabbit polyclonal antibody against α-SMA (Abcam, Cambridge, MA, USA). After washing, biotinylated goat anti-rabbit IgG (1:200, Vector Labs, Burlingame, CA, USA) were applied to the sections for 30 min at room temperature. Lung sections were incubated with Streptavidin-HRP (Vector Labs, Burlingame, CA, USA) for 30 min at room temperature. Diaminobenzidine (DAB; Vector Labs, Burlingame, CA, USA) was used as the chromogen and hematoxylin as the counterstain.

### 4.7. Lung Histology and Histologic Score

Tissue sections from 1 portion of the lung were stained with hematoxylin and eosin (H&E) staining. The H&E-stained sections were examined for lung morphology using ImageJ (National Institutes of Health, Bethesda, MD, USA). A minimum of 10 fields for each lung slide was examined and scored for pathological injury. The average histological score for each sample was calculated based on four histological parameters: linear intercept, septal thickness, area disrupted, and destructive index [67,68]. The images were captured with an Olympus BX51 microscope, 40× magnification (Olympus, Center Valley, PA, USA).

### 4.8. Statistical Analysis

Statistical analyses were performed with GraphPad software (Prism 5.02, San Diego, CA, USA). The quantitative analyses, such as qPCR data, were reported as the mean ± SEM and analyzed using 1- or 2-way ANOVA, followed by a Dunnett’s multiple comparisons test or a Bonferroni post-hoc analysis and unpaired *t*-test. Evaluation of the histopathology preparations was blind, and specimen identity was revealed only after completion of the analyses. A *p*-value of <0.05 was considered to be statistically significant.

## Figures and Tables

**Figure 1 ijms-22-05244-f001:**
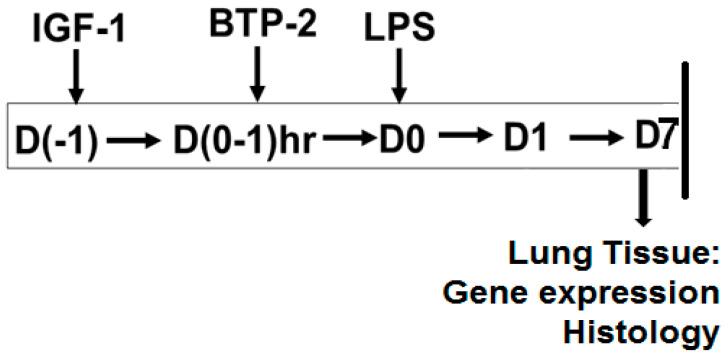
Schematic representation of the experimental setup used for the ALI studies.

**Figure 2 ijms-22-05244-f002:**
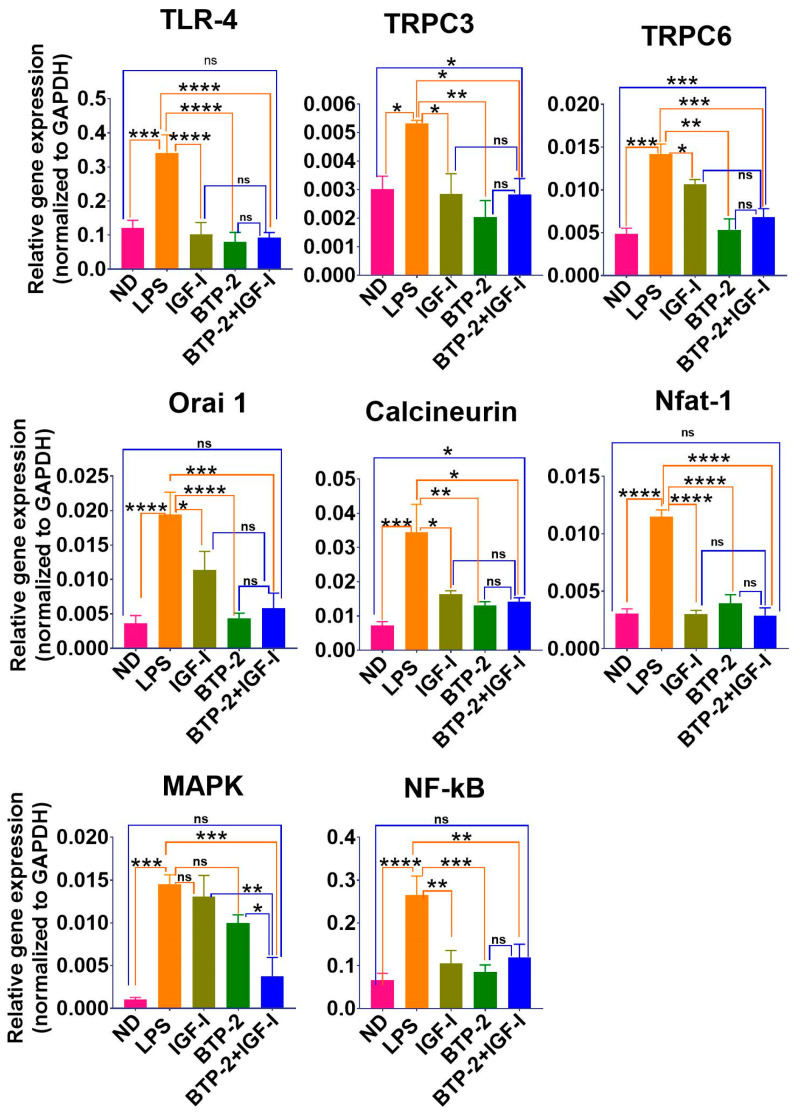
Effects of IGF—I, BTP—2 monotherapies, and IGF—I + BTP—2 combination therapies on the mRNA expression of calcium channel regulators in the lungs. Day 7 RT—qPCR of the markers TLR—4, TRPC3, TRPC6, Orai1, calcineurin, Nfat—1, and downstream signaling markers MAPK and NF—KB. qPCR data are the mean ± SEM. * *p* < 0.05, ** *p* < 0.01, *** *p* < 0.001, and **** *p* < 0.0001, ns-no significance. One-way ANOVA, followed by a Bonferroni’s multiple comparisons test.

**Figure 3 ijms-22-05244-f003:**
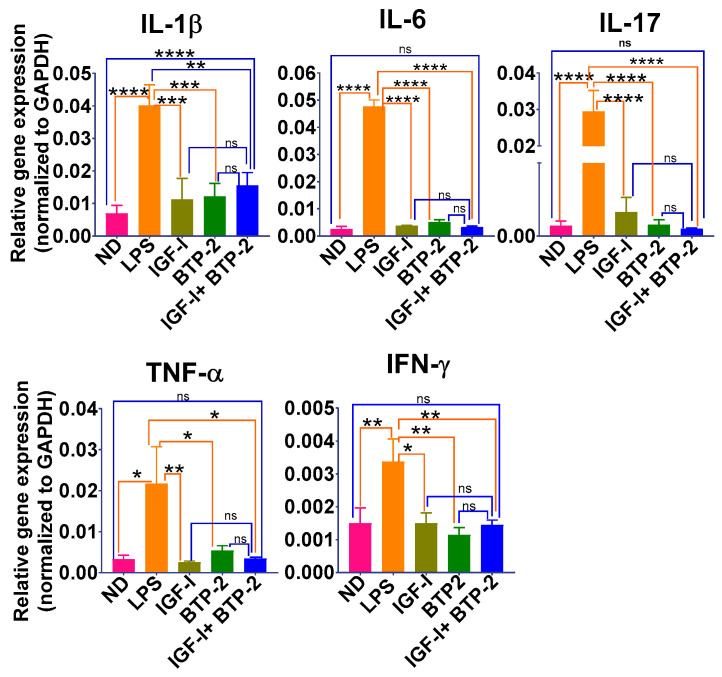
Effects of IGF—I, BTP—2 monotherapies, and IGF—I + BTP—2 combination therapies on the major proinflammatory cytokines in the lungs. Day 7 RT—qPCR of the markers IL—1β, IL—6, IL—17, TNF—α, and IFN—γ. Data are the mean ± SEM. * *p* < 0.05, ** *p* < 0.01, *** *p* < 0.001, and **** *p* < 0.0001, ns-no significance. One-way ANOVA, followed by a Bonferroni’s multiple comparisons test.

**Figure 4 ijms-22-05244-f004:**
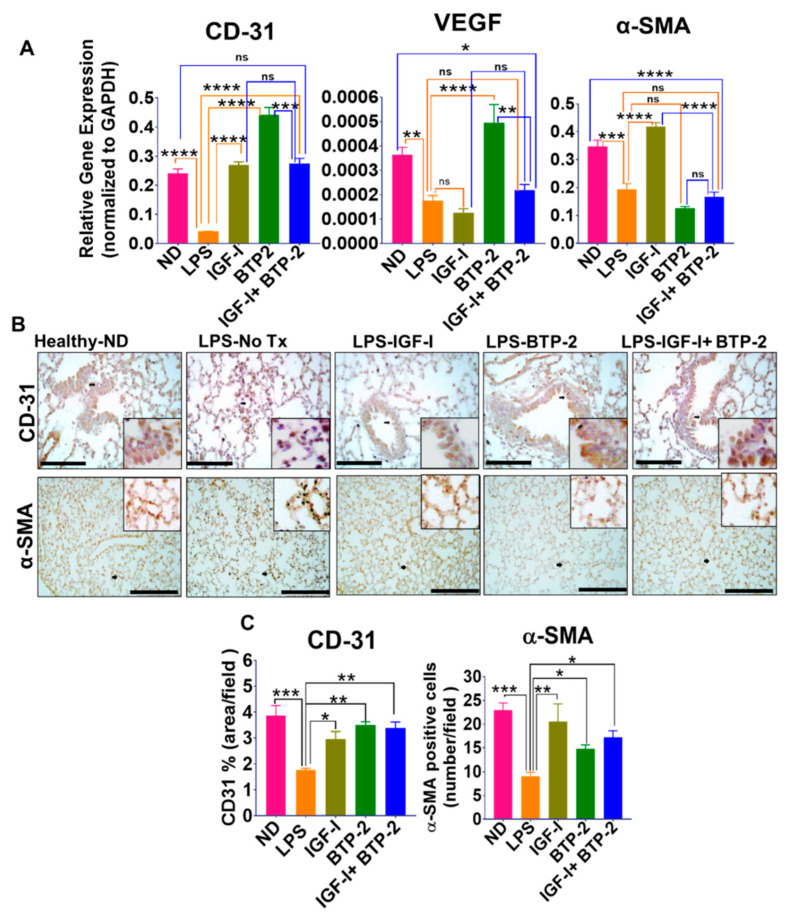
Effects of the IGF—I and BTP—2 monotherapies and the IGF—I + BTP—2 combination therapy on LPS—induced acute lung injury (ALI)—associated vascular integrity markers within the lungs. (**A**) Day 7 RT—qPCR for the vascular markers CD31, VEGF, and α—SMA (**B**) IHC staining for CD31 and α-SMA in the lung sections, and (**C**) a quantitative representation of the IHC data in the form of a % of the area of CD31 staining and the average number of a—SMA—positive cells per field. Data are the mean ± SEM. * *p* < 0.05, ** *p* < 0.01, *** *p* < 0.001, **** *p* < 0.0001 and ns—no significance. One—way ANOVA, followed by Dunnett’s multiple comparisons test. Scale: 200 µM.

**Figure 5 ijms-22-05244-f005:**
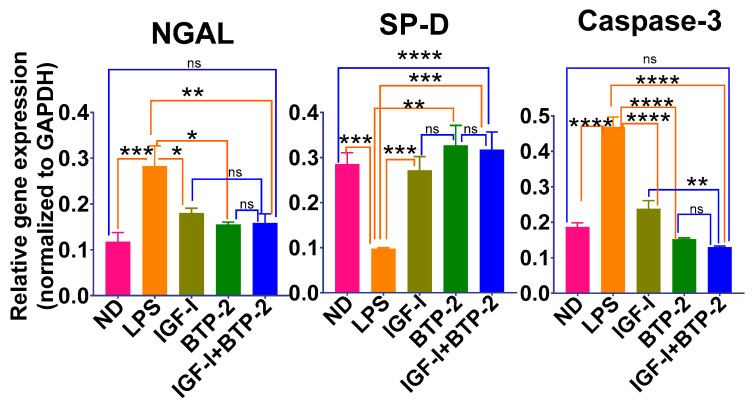
Effects of the IGF—I and BTP—2 monotherapies and the IGF—I + BTP—2 combination therapy on LPS of the injury and repair markers in the lungs. Day 7 RT—qPCR for the injury markers NGAL and caspase-3 and repair marker SP—D. Data are the mean ± SEM. * *p* < 0.05, ** *p* < 0.01, *** *p* < 0.001, **** *p* < 0.0001 and ns—no significance. One-way ANOVA, followed by Bonferroni’s multiple comparisons test.

**Figure 6 ijms-22-05244-f006:**
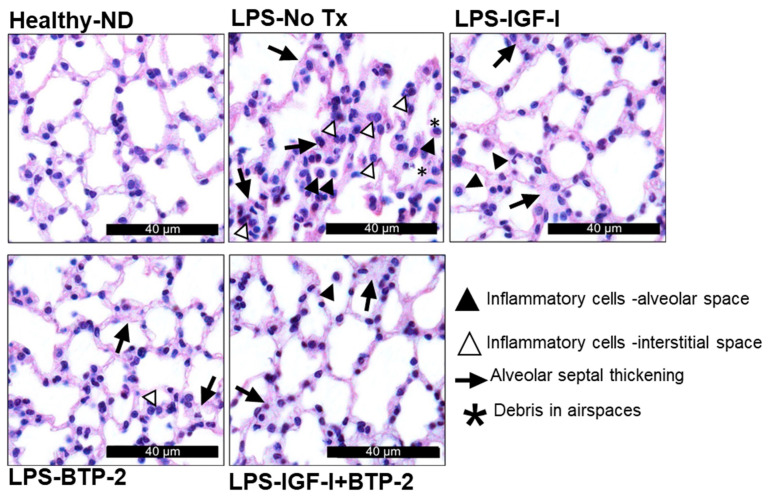
Effects of the IGF—I and BTP—2 monotherapies and the IGF—I + BTP—2 combination therapy on the tissue structures in the lungs. H&E staining of the lung section on Day 7. Scale: 40 µM.

**Figure 7 ijms-22-05244-f007:**
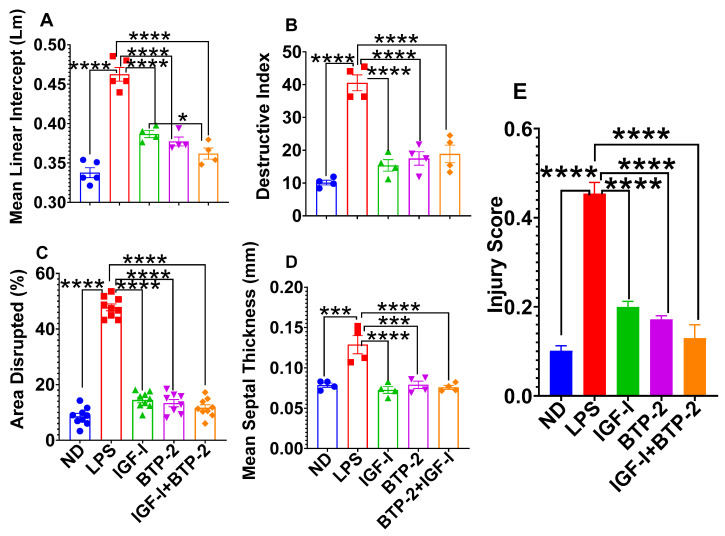
Effects of the IGF—I and BTP—2 monotherapies and the IGF—I + BTP—2 combination therapy on the tissue structures in the lungs. The histomorphometry parameters (**A**) Mean linear intercept, (**B**) destructive index, (**C**) percentage of the area disrupted, (**D**) mean septal thickness, and (**E**) the injury score were quantified by the evaluation on Day 7 of the lung sections stained by H&E. The histology quantification data are the mean ± SEM. * *p* < 0.05, *** *p* < 0.001, and **** *p* < 0.0001. One—way ANOVA, followed by Dunnett’s multiple comparisons test.

## Data Availability

The data is contained within the article.

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
