# Peer review of "The Effects of Insulin-Like Growth Factor I and BTP-2 on Acute Lung Injury"

_ijms, 2021, doi:10.3390/ijms22105244_

Round 1

Reviewer 1 Report

The goal of this study was to develop a therapy to normalize excess production of inflammatory cytokines, and to enhance tissue repair in mice exposed to LPS. The growth factor IGF-1 and the Orai  Ca2+ channel inhibitor BTP-2 were tested separately and in combination. The results showed that LPS challenge stimulated mRNA expression of inflammatory cytokines including IL-1β, IL-17, TNF-α, and IFN- γ as well as TLR4, and various calcium channel regulators. In addition, IGF-1+BTP-2 dual therapy improved vascular gene expression markers. LPS significantly increased the histologic lung injury score, which was reduced in mice receiving combination therapy. The authors conclude that LPS-induced increases in inflammatory cytokines, vascular injury, and ALI were all improved by the IGF-I + BTP-2 combination therapy.

Overall, the results of the study support the conclusions but I have some minor concerns that must be addressed.

Comments:

  1. Figures legends 2-5 and 7: Although an ANOVA is appropriate, it is worth noting that Dunnett’s method only allows for multiple comparisons with a common control condition. In these figures multiple comparisons are being made between all treatment conditions, so I believe that the author’s meant to say that a Bonferroni post-test was used (based on tests identified in the Methods under Statistical Analysis), not Dunnett’s test.
  2. (Fig. 5) I found it interesting that LPS induced expression of caspase 3. Was this increase in mRNA expression associated with increased caspase 3 catalytic activity? If so, was there evidence for apoptosis associated with LPS exposure and was this blocked by IGF-1, BTP-2 or in combination?
  3. Lines 303-305: “This action on cell death would be expected to release damage-associated molecular pattern (DAMPs), which would promote a sterile inflammatory response and the infectious inflammatory response of the virus.” What virus are the authors referring too? My understanding is that only LPS was used to induce inflammation and ALI. Are the authors implying an effect of the lentivirus vector used to overexpress IGF-1?
  4. Lines 360-363: “Therefore, it is reasoned that IGF-I treatment is mediated, at least in part, through regulatory effects on cytosolic calcium. A thorough study will be required to explain the potential additional effects of IGF-I on intracellular calcium regulation and its functional consequences.” The authors should consider including the following reference. Although the study focuses on skeletal muscle, it is possible (and perhaps likely) that a similar response to IGF-1 occurs in lung tissue.

Valdes et al., IGF-1 induces IP3 -dependent calcium signal involved in the regulation of myostatin gene expression mediated by NFAT during myoblast differentiation, J. Cell Physiol. (2013) 228(7):1452-63. doi: 10.1002/jcp.24298. 

Author Response

Thank you for allowing us to revise our manuscript, “The Effects of Insulin-like Growth Factor I and BTP-2 on Acute Lung Injury”. We appreciate the time and effort that the reviewers have dedicated to providing feedback on our manuscript. The feedback was very insightful and valuable for improving the quality of our paper.

We have incorporated most of the suggestions made by the reviewers. Those changes are highlighted in the main text of the manuscript. Please also see below for point-by-point responses to the reviewers’ comments and concerns.

Reviewer 1

  1. Figures legends 2-5 and 7: Although an ANOVA is appropriate, it is worth noting that Dunnett’s method only allows for multiple comparisons with a common control condition. In these figures multiple comparisons are being made between all treatment conditions, so I believe that the author’s meant to say that a Bonferroni post-test was used (based on tests identified in the Methods under Statistical Analysis), not Dunnett’s test.

Response: Thank you for the correction. Your statistical point is well taken. We have now replaced Dunnett’s test with Bonferroni's post-test in Figures 2, 5, and 7.

  1. (Fig. 5) I found it interesting that LPS induced expression of caspase 3. Was this increase in mRNA expression associated with increased caspase 3 catalytic activity? If so, was there evidence for apoptosis associated with LPS exposure and was this blocked by IGF-1, BTP-2 or in combination?

Response: We did not measure caspase 3 catalytic activity or apoptosis; however, LPS is known to cause endothelial cell apoptosis [Munshi, N et al, Haimovitz-Friedman, A et al, Kawasaki, M et al]. We have now indicated this potential result in the main text. Lines-226-228.

Munshi, N., et al., Lipopolysaccharide-induced apoptosis of endothelial cells and its inhibition by vascular endothelial growth factor. J Immunol, 2002. 168(11): p. 5860-6.

Haimovitz-Friedman, A., et al., Lipopolysaccharide induces disseminated endothelial apoptosis requiring ceramide generation. J Exp Med, 1997. 186(11): p. 1831-41.

Kawasaki, M., et al., Protection from lethal apoptosis in lipopolysaccharide-induced acute lung injury in mice by a caspase inhibitor. Am J Pathol, 2000. 157(2): p. 597-603.

  1. Lines 303-305: “This action on cell death would be expected to release damage-associated molecular pattern (DAMPs), which would promote a sterile inflammatory response and the infectious inflammatory response of the virus.” What virus are the authors referring too? My understanding is that only LPS was used to induce inflammation and ALI. Are the authors implying an effect of the lentivirus vector used to overexpress IGF-1?

Response: Thank you for noticing our error. We have now changed virus to microbe in the main text. Line- 293.

  1. Lines 360-363: “Therefore, it is reasoned that IGF-I treatment is mediated, at least in part, through regulatory effects on cytosolic calcium. A thorough study will be required to explain the potential additional effects of IGF-I on intracellular calcium regulation and its functional consequences.”The authors should consider including the following reference. Although the study focuses on skeletal muscle, it is possible (and perhaps likely) that a similar response to IGF-1 occurs in lung tissue.

Valdes et al., IGF-1 induces IP3 -dependent calcium signal involved in the regulation of myostatin gene expression mediated by NFAT during myoblast differentiation, J. Cell Physiol. (2013) 228(7):1452-63. doi: 10.1002/jcp.24298. 

Response: This is a good reference. We have now added to the text along with the reference that “IGF-I can induce IP 3 in skeletal muscle [Valdes, J.A et al] and therefore could release ER calcium into the cytosol, an action which could increase cytosolic calcium. Therefore, further studies are required to understand the complex actions of IGF-I in the context of our LPS model”. Lines 343-345.

Reviewer 2 Report

Overexpression of inflammatory cytokines such as TNF-α, IL-1β and others is responsible for the pathogenesis of acute lung injury (ALI). In this manuscript, Munoz et al. found that IGF-1, combined with BTP-2, significantly suppressed the expression of those proinflammatory genes and certain ALI biomarkers in the lung of mice treated with LPS and improved the recovery of lung injury. Overall, the findings are very interesting and the manuscript was well written.

Major concerns:

  1. LPS activates the TLR4 signaling pathway, leading to the expression of proinflammatory genes, resulting in lung tissue damage. In this study, the authors measured the RNA level of TLR4 in the lung tissues after mice were treated with LPS. How about the TLR4 protein? Most importantly, TLR4 must be phosphorylated to be activated. Thus, it is necessary to perform ELISA for determining the TLR4 protein level and examine the activation of TLR4 by Western blot for p-TLR4.
  2. Activation of TLR4 initiates a cascade of signaling. Although a variety of biomarkers were up-regulated at their RNA levels as determined by RT-PCR, the activation of the down-stream signaling pathways such as MAPK and IkB/NFkB should be examined as well.

Author Response

Thank you for allowing us to revise our manuscript, “The Effects of Insulin-like Growth Factor I and BTP-2 on Acute Lung Injury”. We appreciate the time and effort that the reviewers have dedicated to providing feedback on our manuscript. The feedback was very insightful and valuable for improving the quality of our paper.

We have incorporated most of the suggestions made by the reviewers. Those changes are highlighted in the main text of the manuscript. Please also see below for point-by-point responses to the reviewers’ comments and concerns.

Reviewer 2

Major concerns:

  1. LPS activates the TLR4 signaling pathway, leading to the expression of proinflammatory genes, resulting in lung tissue damage. In this study, the authors measured the RNA level of TLR4 in the lung tissues after mice were treated with LPS. How about the TLR4 protein? Most importantly, TLR4 must be phosphorylated to be activated. Thus, it is necessary to perform ELISA for determining the TLR4 protein level and examine the activation of TLR4 by Western blot for p-TLR4.

Response: We agree that TLR-4 is pivotal in the action of LPS. We have no sample remaining to measure TLR-4 by Western blot. We have, however, cited previously published literature showing that LPS initiates its activity by activating the TLR-4-mediated intracellular signaling pathway [22,29]. In addition, it also increases the expression of TLR-4 protein [Wang, P et al.]. Lines-124-125.

Wang, P., et al., LPS enhances TLR4 expression and IFNgamma production via the TLR4/IRAK/NFkappaB signaling pathway in rat pulmonary arterial smooth muscle cells. Mol Med Rep, 2017. 16(3): p. 3111-3116.

  1. Activation of TLR4 initiates a cascade of signaling. Although a variety of biomarkers were up-regulated at their RNA levels as determined by RT-PCR, the activation of the down-stream signaling pathways such as MAPK and IkB/NFkB should be examined as well.

Response: This is a good recommendation. Accordingly, we have now measured MAPK and NF-kB, the results of which are now added to the text, along with the new dataset and reference [Kuper, Beck et al.]. Lines-136-138 and 148-149.

Kuper, C., F.X. Beck, and W. Neuhofer, Toll-like receptor 4 activates NF-kappaB and MAP kinase pathways to regulate expression of proinflammatory COX-2 in renal medullary collecting duct cells. Am J Physiol Renal Physiol, 2012. 302(1): p. F38-46.

Round 2

Reviewer 2 Report

None